# StyleCL: Latent Dictionary Learning for StyleGAN without Forgetting

## Abstract

StyleGAN is one of the most versatile generative models that have emerged in recent times. However, when it is trained continually on a stream of data (potentially previously unseen distributions), it tends to forget the distribution it has learned, as is the case with any other generative model, due to catastrophic forgetting. Recent studies have shown that the latent space of StyleGAN is very versatile, as data from a variety of distributions can be inverted onto it. In this paper, we propose to leverage this property to facilitate lifelong learning of StyleGAN without forgetting. Specifically, given a StyleGAN trained on a certain task (dataset), we propose to learn a set of dictionary vectors in its latent space, one for each novel, unseen task (or dataset). Additionally, we also learn a relatively small set of shared parameters (feature adaptors) in the weight space to complement the dictionary learning in the latent space. During inference, given a dataset/task, our method invokes the corresponding learned latent dictionary and the shared parameters for that particular task. Our method avoids catastrophic forgetting because the set of dictionary and the feature adaptor parameters are unique for each task. However, the generator for each task shares all of the parameters except for the newly added parameters of the feature adaptor. We demonstrate that our method, StyleCL, achieves better generation quality on multiple datasets. Additionally, our method requires significantly fewer additional parameters per task compared to previous methods. This is a consequence of learning task-specific dictionaries in the latent space, which has a much lower dimensionality compared to the weight space. We also demonstrate that our method, StyleCL, offers the capability for positive forward transfer for semantically similar tasks.

## 1 Introduction

Continual learning (CL) is a fundamental machine learning paradigm that focuses on the model's ability to learn and adapt to new tasks or evolving data streams over time while ensuring that previously acquired knowledge remains intact. Extensive research has explored continual learning within the context of discriminative models De Lange et al. (2022), but relatively less attention has been devoted to the application of this paradigm in the realm of generative models. However, recent progress in the field of generative modelling has brought them to the forefront of application domains. Specifically, models such as Generative Adversarial Networks (GANs) Goodfellow et al. (2014) and denoising diffusion models Ho et al. (2020) have found their utility in a wide variety of tasks such as semantic editing Ling et al. (2021), image in-painting, Yu et al. (2018) etc. Thus, it is imperative to consider the problem of continual learning in the context of generative models Lesort et al. (2019).

In particular, we direct our attention to continual learning in StyleGAN Karras et al. (2020b) which is one of the most popular variants of GANs. We hypothesize, that StyleGAN is suited for such cases because of its versatility, in that, a large variety of datasets can be inverted onto its extended latent space $(\mathcal{W}^+)$ as observed in Abdal et al. (2019). Motivated by these observations, we investigate whether the latent space of StyleGAN can be exploited to generate data from a stream of datasets without forgetting. Towards that end, we propose a method to learn a per-task, style-wise dictionary of vectors that define a subspace in the latent space of StyleGAN. In addition to latent dictionary learning, we also learn a set of shared parameters in the weight space, to accommodate a richer knowledge in tandem with the learned latent subspace.

Knowledge transfer, a cornerstone of continual learning, assumes a central role in StyleCL. StyleCL utilises the latent space to identify the most similar task unlike GAN Memory Cong et al. (2020) and CAM-GAN Varshney et al. (2021) where the most similar task is characterized using the most recent task or the task with high Fisher information respectively. We also determine the nature of forward knowledge transfer (positive or negative) by measuring the cosine similarity of dictionary vectors to its projection onto the latent subspace of the most similar task which is then used to prevent negative forward transfer.

Moreover, we expand the scope of generative continual learning to encompass real-world scenarios where data from multiple tasks arrive simultaneously without task identification (task ID). Notably, StyleCL adeptly extends its applicability to such settings with minimal adjustments to its training strategy. StyleCL accomplishes this by segregating distinct tasks into distinct regions in the latent space. Even in these scenarios where supervision on task ID is not available, StyleCL consistently delivers high-quality generation capabilities.

The following is a summary of our contributions:

- **Latent subspace learning for StyleGAN**: We propose a latent subspace learning approach that enables learning without forgetting for StyleGAN.

- **Improved generation quality**: By harnessing the versatility of StyleGAN's latent space, our method outperforms contemporary approaches like CAM-GAN and GAN Memory in terms of generation quality, all while employing fewer parameters (28.95% reduction) and FLOPs (11.6% reduction).

- **Prevention of negative forward transfer**: We further propose a simple way to identify the most similar previous task and also characterize the nature of forward transfer between any two tasks to prevent negative forward transfer.

- **Extension to task ID free setting**: We extend StyleCL to scenarios where task ID is not available wherein StyleCL discovers different data distributions automatically.

## 2 RELATED WORK

**Generative Continual Learning**: Continual Learning methods are broadly categorized into three categories: replay-based, regularization-based and parameter isolation-based methods. These categorizations are defined for a discriminative continual setting but they can be applied to generative continual learning as well.

Chenshen et al. (2018) introduces MerGAN, a replay-based GAN that combines generated samples from previous tasks with new task data to form an extended training dataset. They also introduce a replay-alignment loss to ensure consistent generation for previous tasks as the number of tasks increases. Zhai et al. (2019) presents Lifelong GAN for continual image-conditioned image generation, employing knowledge distillation and auxiliary data generation by creating patch montages from training batches to mitigate catastrophic forgetting. However, replay-based approaches face scalability issues due to cumulative inaccuracies when a single generator is incrementally updated.

Parameter isolation techniques like PiggybackGAN Zhai et al. (2020) freeze old task parameters and introduce smaller new parameters for learning without forgetting. GAN memory Cong et al. (2020) employs normalization parameters to adapt the generator's weights to incoming data streams. CAM-GAN Varshney et al. (2021) introduces adaptation modules via group-wise convolutions at the output of each convolution layer in the base network. In contrast, StyleCL takes a different approach by learning a latent subspace alongside shared weight space parameters, facilitating continual learning.

Even though few regularisation-based approaches like Liang et al. (2018) and Seff et al. (2017) use regularisations to enable continual learning, their generation quality still degrades over time and thus parameter isolation methods appear to be a better choice and have been receiving more attention.

**Knowledge transfer in continual learning**: Knowledge transfer is a crucial aspect of continual learning, predicated on the notion that similar tasks inherently possess shared knowledge that can be effectively transferred between them. However, previous approaches, like MerGAN Chenshen et al. (2018), Lifelong GAN Zhai et al. (2019), and Piggyback GAN Zhai et al. (2020), often lack explicit mechanisms to facilitate this positive knowledge transfer. While GAN Memory Cong et al. (2020)

demonstrates promise in enabling knowledge transfer, it relies on the assumption that the most recent task is invariably the most similar, a notion that does not consistently hold. In contrast, CAM-GAN Varshney et al. (2021) quantifies task similarity by approximating the Fischer information matrix (FIM) and posits that initializing the current task with parameters from the most similar task would consistently yields positive forward transfer which may not always hold true. StyleCL distinguishes itself by characterizing both the most similar task and the nature of forward transfer using the latent space, thus effectively capturing the state of the generator while identifying the most similar task and elucidating the nature of the forward knowledge transfer.

**Continual Learning beyond GANs**: Continual learning is a dynamic field that extends beyond GANs. While Variational Autoencoders (VAEs) have been considered in the past, their subpar generation quality has led to a recent decline in attention. In contrast, the exploration of continual learning in Diffusion models represents an emerging paradigm. Recent studies, such as those by Gao & Liu (2023) and Chen et al. (2023), have delved into the utility of diffusion models for replaying previous data in the context of discriminative continual learning.

It is essential to note that Diffusion models offer remarkable generation quality enhancements, albeit with a trade-off of increased inference time. On the other hand, GANs excel in efficiency, requiring only a single forward pass. Furthermore, the introduction of GigaGAN, Kang et al. (2023) and StyleGAN-T Sauer et al. (2023) have illustrated their ability to provide competitive generation quality while maintaining faster inference speeds. Given these advantages, we turn our attention to continual learning in GANs, leveraging their rapid inference capabilities while upholding competitive performance compared to other generative models. Additionally, it is worth highlighting that many recent state-of-the-art GANs for various tasks, as seen in works like Kang et al. (2023), Sauer et al. (2023), and Fu et al. (2022), employ StyleGAN- based architectures. This inspires our investigation into StyleGAN-based architectures for continual learning.

## 3 PROPOSED METHOD: STYLECL

### 3.1 PROBLEM SETTING AND METHOD OVERVIEW

Our setting is that a stream of datasets (or tasks) arrive sequentially with a unique task ID. We assume that at any given time, only one dataset is available for training. Formally, let $\{\mathcal{X}^t\}_{t=1}^T$ denote the sequential stream of datasets where $\mathcal{X}^t = \{\mathbf{x}_j^t\}_{j=1}^N$, $\mathcal{X}^t \sim p_t(\mathbf{x}^t)$, with $\mathbf{x}_j^t$ denoting the $j^{th}$ instance from the $t^{th}$ task/dataset. The objective is to train a GAN that can sample from the current dataset without forgetting to sample from all the previously seen $t-1$ distributions.

Our method starts by training a GAN as in Karras et al. (2020a) on the first (or base) dataset (task), denoted by $\mathcal{G}^1$. The parameters of $\mathcal{G}^1$ are denoted by $\phi^1$ and are shared by all the subsequent tasks. For each dataset $\mathcal{X}^t$, our method first selects the most similar previous task and the corresponding generator $\mathcal{G}^k$ and learn the following components to obtain $\mathcal{G}^t$: (i) A set of dictionary vectors $\mathbf{U}^t$ on the latent space of $\mathcal{G}^k$ and (ii) a set of feature adaptor blocks $\phi^t$ ($1 \times 1$ convolutions) on the weight space of $\mathcal{G}^k$. To maintain simplicity, we make the assumption throughout this paper that the feature adaptor $\phi^k$ of generator $\mathcal{G}^k$ encompasses $\phi^1$, the feature adaptor of task $k$, and the feature adaptors of the most similar previous tasks of task $k$. It is noteworthy that in our method, the dictionary vectors are unique for each task, whereas the feature adaptors are shared and added recursively based on the similarity of tasks. Specifically, every $\mathcal{G}^t$ comprises $\phi^1$, the feature adaptor block of its own as well as the feature adaptor block of the most similar previous tasks. Fig. 1 presents an overview of our method, which we name 'StyleCL'.

### 3.2 LATENT DICTIONARY LEARNING

We employ StyleGAN2 Karras et al. (2020a) architecture for the generators $\mathcal{G}$ that contains $M$ style blocks and for simplicity of discussion, we assume each of these style blocks comprises of just 1 layer. The first stage of our method is to learn a set of dictionaries on the extended latent space ($\mathcal{W}^+$ space) of the StyleGAN. Given a dataset $\mathcal{X}^t$, a dictionary $\mathbf{U}_m^t = \{\mathbf{u}_{m1}^t, \ldots, \mathbf{u}_{mi}^t, \ldots, \mathbf{u}_{mK}^t\}, \mathbf{u}_{mi}^t \in \mathbb{R}^d$, containing $K$ vectors are learned for each of the $m = 1, 2, .., M$ style blocks of the generator.

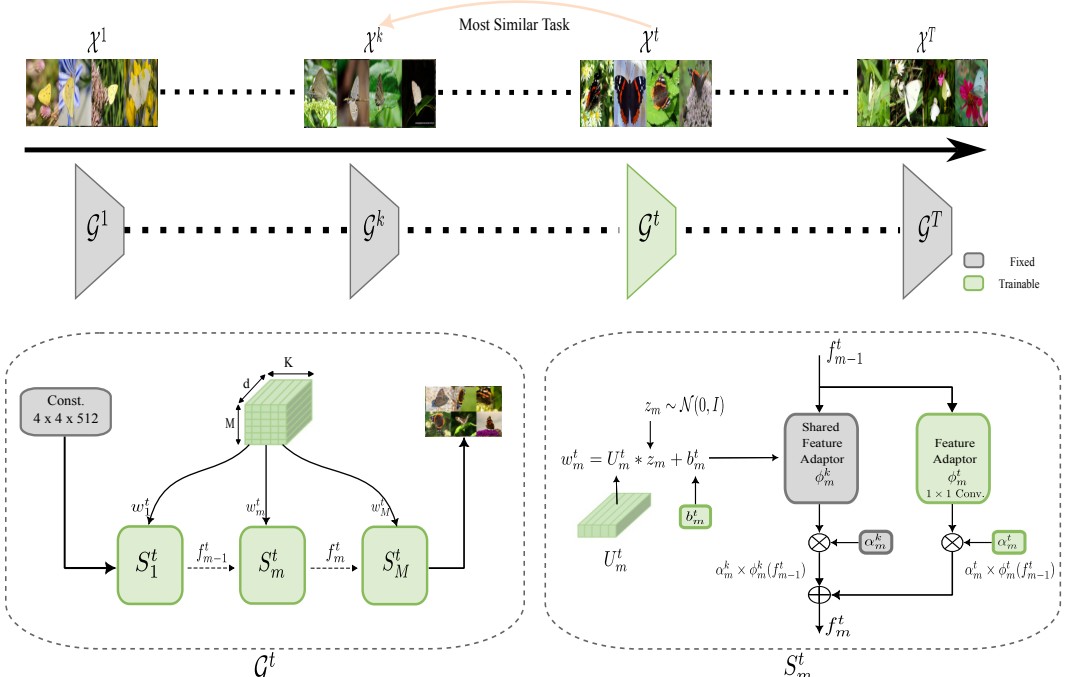

Figure 1: Overview of StyleCL: Given a dataset $\mathcal{X}^t$ at time $t$, the most similar previous generator $\mathcal{G}^k$ is first selected as the base generator. A set of $K$ dictionary vectors, each for a style block $m$, is learned for $\mathcal{X}^t$. Further, a feature adaptor block $\phi_m^t$ is added to the existing shared feature adaptor block $\phi_m^k$ in $\mathcal{G}^k$. During inference, sampling from $\mathcal{X}^t$ is done by giving stochastic combinations of elements of the corresponding dictionary vectors as input to $\mathcal{G}^t$.

During the training, the parameters of $\mathbf{U}_m^t$ are initialized randomly. First, a batch of vectors is stochastically sampled from each dictionary $\mathbf{U}_m^t$ as follows:

$$\mathbf{w}_m^t = z_{m1}\,\mathbf{u}_{m1}^t + z_{m2}\,\mathbf{u}_{m2}^t \ldots + z_{mK}\,\mathbf{u}_{mK}^t + \mathbf{b}_m^t \tag{1}$$

where $\mathbf{z}_m = [z_{m1}, \ldots, z_{mK}] \sim \mathcal{N}(0, \mathbf{I})$ and $\mathbf{b}_m^t$ is the bias term. Further, each $\mathbf{w}_m^t$ corresponding to every style block is concatenated to form $\mathbf{w}^t$ as:

$$\mathbf{w}^t = [\mathbf{w}_1^t, \ldots, \mathbf{w}_M^t], \mathbf{w}^t \in \mathcal{W}^+ \tag{2}$$

Finally, $\mathbf{w}^t$ is passed as the input to the fixed generator $\mathcal{G}^k$ obtained from the most similar task to generate images from $p_t(\mathbf{x}^t)$.

### 3.3 FEATURE ADAPTORS IN THE WEIGHT SPACE

We observed empirically that the ability of the latent dictionary to capture a distribution $p_t(\mathbf{x}^t)$, depends on $\mathcal{G}^k$. Therefore, learning $\mathbf{U}_m^t$ alone may not be fully sufficient to model $p_t(\mathbf{x}^t)$. Therefore we also introduce additional feature adaptor blocks on the weight space of the generator of the most similar task $\mathcal{G}^k$ to obtain $\mathcal{G}^t$. Since the latent subspace would have already captured some characteristics of the datasets, the number of feature adaptor parameters to be learned would be lesser (Tab. 1).

Let $S_m^t$ denote the $m^{th}$ style block within the generator $\mathcal{G}^t$ for task $t$. Initially, we identify the most similar task to $t^{th}$ task which is denoted by $k$ and the corresponding generator $\mathcal{G}^k$ is selected as the base generator. We introduce a trainable feature adaptor block $\phi_m^t$ ($1 \times 1$ Convolution layer) to the existing shared feature adaptor block $\phi_m^k$ in $\mathcal{G}^k$ to obtain $S_m^t$ of $\mathcal{G}^t$. When the $t^{th}$ task emerges, we learn $\phi_m^t$ and compute the new activation map of $S_m^t$ as follows:

$$f_m^t = \alpha_m^k \times \phi_m^k(f_{m-1}^t) + \alpha_m^t \times \phi_m^t(f_{m-1}^t) \tag{3}$$

The feature adaptor block $\phi_m^t$ is intended to learn additional information that is absent in $\mathcal{G}^k$. Here both $\phi_m^t$ and scaling coefficient $\alpha_m^t$ are learnable and jointly learned with the latent dictionary. It

is important to note that $\phi_m^k$ is shared between $\mathcal{G}^k$ and $\mathcal{G}^t$, whereas $\phi_m^t$ is exclusively for $\mathcal{G}^t$. We follow the training paradigm in Karras et al. (2020a), which includes adversarial loss $\mathcal{L}_1$ and to ensure smoothness and facilitate better convergence, we use the Perceptual Path Regularizer (PPL) Karras et al. (2020b) and $\mathcal{R}1$ regularization Mescheder et al. (2018) as in Karras et al. (2020b).

---

**Algorithm 1** StyleCL : Training

---

**Input:** $\{\mathcal{X}^t\}_{t=1}^T$: Sequential Data stream, where $T$ is the total number of tasks
**Output:** $\{\mathbf{U}^t\}_{t=2}^T$: where $\mathbf{U}^t = \{\mathbf{U}_m^t\}_{m=1}^M$; $\{\mathbf{b}^t\}_{t=2}^T$; $\mathcal{G}^1$; $\{\phi^t\}_{t=2}^T$ where $\phi^t = \{\phi_m^t\}_{m=1}^M$
1: Train StyleGAN2 on $\mathcal{X}^1$ to obtain $\mathcal{G}^1$
2: **for** $t = 2 \dots T$ **do**
3:     Initialize Discriminator parameters $\psi$, and set of dictionary vectors $\mathbf{U}^t$ and $\mathbf{b}^t$
4:     Find the most similar previous task $k$ using Eq. (6) to obtain $\mathcal{G}^k$
5:     **for** each training iteration **do**
6:         Obtain $\mathbf{w}^t$ using Eq. (1) and Eq. (2)
7:         Optimize parameters $\mathbf{U}^t$, $\mathbf{b}^t$ using Eq. (4) combined with PPL and $\mathcal{R}1$ regularization

$$\mathcal{L}_1 = \mathbb{E}_{x \sim p_t(\mathbf{x}^t)}[\log D_\psi(x)] + \mathbb{E}_{\mathbf{z}_1,\dots\mathbf{z}_M \sim \mathcal{N}(0,\mathbf{I})}[1 - \log D_\psi(\mathcal{G}^k(\mathbf{w}^t))] \tag{4}$$

8:     **end for**
9:     Compute $sim(t,k)$ using Eq. (7)
10:    Initialize parameters $\phi_t, \alpha_t$
11:    **if** $sim(t,k) > 0$ **then**
12:        $\mathcal{G}^t = \mathcal{G}^k \cup \phi^t$
13:    **else**
14:        $\mathcal{G}^t = \mathcal{G}^1 \cup \phi^t$
15:    **end if**
16:    Optimize parameters $\mathbf{U}^t$, $\mathbf{b}^t$ and $\phi^t$ using Eq. (5) combined with PPL and $\mathcal{R}1$ regularization

$$\mathcal{L}_2 = \mathbb{E}_{x \sim p_t(\mathbf{x}^t)}[\log D_\psi(x)] + \mathbb{E}_{\mathbf{z}_1,\dots\mathbf{z}_M \sim \mathcal{N}(0,\mathbf{I})}[1 - \log D_\psi(\mathcal{G}^t(\mathbf{w}^t))] \tag{5}$$

17: **end for**

---

### 3.4 FORWARD TRANSFER: CHOOSING THE MOST SIMILAR PREVIOUS TASK

Given a task $t$, our method first chooses the generator $\mathcal{G}^k$ of the most similar task and learns feature adaptors over it. This is akin to the idea of forward transfer in the continual learning literature Chen & Liu (2018). The dictionary vectors learned for each task allow easy characterisation of the task and we use these to find the most similar task.

In order to find the task that is most similar to an incoming task, we need to characterize both the previous task as well as the current task in the latent space. We characterize the current task by learning the dictionary vectors alone using the base generator $\mathcal{G}^1$. It is to be noted that the dictionary vectors are already learnt for previous tasks from 2 to $t-1$. Given any task $t$, we use the set of bias vectors learned as task embedding, $\mathbf{b}^t = [\mathbf{b}_1^t, \dots, \mathbf{b}_M^t]$, $\mathbf{b}^t \in \mathcal{W}^+$ since it captures the relative position of the learned latent subspace in the $\mathcal{W}^+$ space. Given the task embedding for the current and previous tasks, we define the most similar task $k$ as the one whose embedding has the least Euclidean distance from the embedding of the current task as provided in Eq. (6). This is motivated by the fact that the latent vectors of similar tasks lie close together while being distant from dissimilar tasks as observed in Fig. 4.

$$k = \underset{r:r \in \{2,\dots,t-1\}}{\operatorname{argmin}} \|\mathbf{b}^t - \mathbf{b}^r\|_2 \tag{6}$$

**Preventing negative forward transfer**: Choosing the most similar task facilitates selecting a task with a similar set of features as that of the current task, however, it may lead to negative forward transfer. In order to alleviate this problem, we estimate the nature of forward transfer (positive or negative) by computing the cosine similarity of dictionary vectors of the current task to its projection onto the latent subspace of the most similar task $k$. Let $\mathbf{V}^k$ correspond to the orthonormal vectors obtained using the Gram-Schmidt orthogonalisation procedure on the dictionary vectors $\mathbf{U}^k$. The projection of $\mathbf{U}^t$ onto the latent subspace characterized by the orthonormal vectors $\mathbf{V}^k$ is then defined

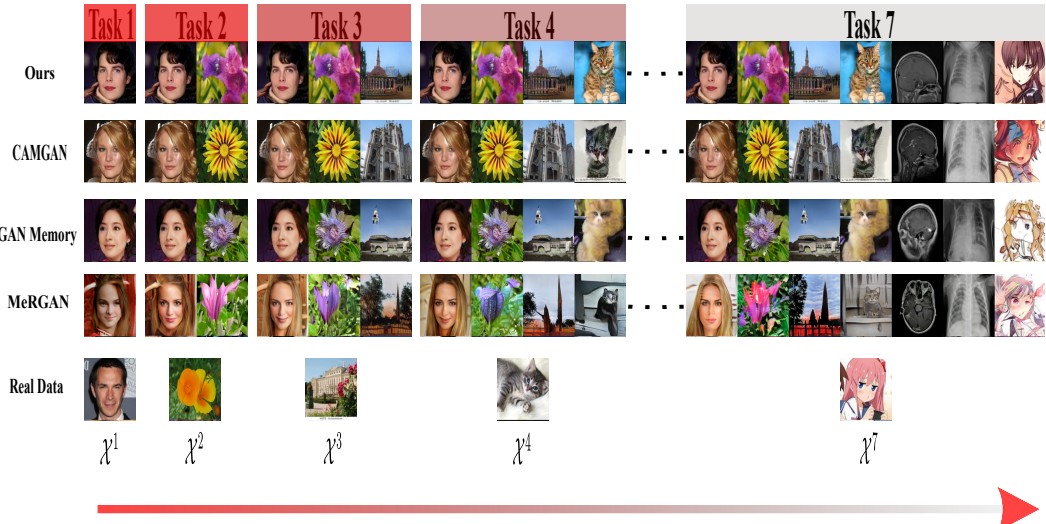

Figure 2: Qualitative results of StyleCL for all six tasks including the base task. Each row corresponds to different baseline methods and each column consists of generated samples from all the tasks till the current instant.

by $\mathbf{V}_i^{k^*}\mathbf{U}_i^t\mathbf{V}_i^k$. Subsequently, we define the nature of forward transfer as follows :

$$sim(t,k) = \frac{1}{M}\sum_{m=1}^{M}\frac{\mathbf{U}_m^t \cdot \left(\mathbf{V}_m^{k^*}\mathbf{U}_m^t\mathbf{V}_m^k\right)}{\left\|\mathbf{U}_m^t\right\|\left\|\mathbf{V}_m^{k^*}\mathbf{U}_m^t\mathbf{V}_m^k\right\|} \qquad (7)$$

$sim(t,k) \in [-1,1]$, and when $sim(t,k) \leq 0$, it signifies zero or negative forward transfer. In such cases, we avoid reusing the parameters of the most similar task, thus preventing negative forward transfer. We provide a comprehensive overview of the StyleCL training process in Algorithm 1.

### 3.5 Overcoming Task ID Constraints with StyleCL

The previously defined problem setting assumes a sequential arrival of datasets (or tasks) with unique task IDs. However, in real-life scenarios, this assumption may not always hold. For instance, data collection from multiple sources can occur simultaneously, resulting in a set of data without task distinction. Specifically, we address scenarios where task $t$ comprises contributions from $Q$ datasets, and task ID is not available. We operate under the assumption that $Q$ is known apriori. We demonstrate that StyleCL can be seamlessly extended to accommodate these scenarios. We initialize $Q$ sets of dictionary vectors and feature adaptors. The intuition is that this modelling choice forces each of the $q$ latent adaptors to concentrate on latent vectors originating from separate regions in the latent space. As a result, the model would effectively allocate distinct data sources to distinct dictionaries. During training, we exclusively employ the $q^{th}$ feature adaptor set for latent vectors generated from $q^{th}$ dictionary where $q \in \{1, 2, \ldots, Q\}$. This ensures that $q^{th}$ feature adaptors capture task-specific knowledge exclusively for $q^{th}$ dataset.

## 4 Experiments and Results

### 4.1 Experimental settings

We conducted three experiments to evaluate our method's effectiveness. First, we tested its ability to generate from perceptually distinct datasets. Second, we assessed knowledge transfer across similar tasks, using six butterfly categories from ImageNet Russakovsky et al. (2015). Third, we demonstrated StyleCL's effectiveness in scenarios without task ID information

## 4.2 BASELINES AND METRICS

We employ StyleGAN2 Karras et al. (2020a) as the base architecture for all experiments. StyleCL is compared to GAN Memory, CAM-GAN with task similarity learning, and MerGAN. Evaluation metrics include Fréchet inception distance (FID) Heusel et al. (2017), Density, and Coverage Naeem et al. (2020). We also consider computational and memory overhead during inference, measured in FLOPs and parameter count Dehghani et al. (2022), crucial factors for CL scalability.

## 4.3 RESULTS FOR PERCEPTUALLY DISTANT TASKS

Following the experimental setup used in CAM-GAN and GAN-Memory, we begin by training a GAN on CelebA-HQ Karras et al. (2018) dataset and then consider a stream of six perceptually distinct datasets, namely Oxford 102 Flowers Nilsback & Zisserman (2008), LSUN Church Yu et al. (2015), LSUN Cats Yu et al. (2015), Brain MRI Cheng et al. (2016), Chest X-Ray Kermany et al. (2018), and Anime Faces [1]. Samples of generated data from all the methods considered can be found in Fig. 2, and it can be observed that StyleCL produces higher-quality generated images.

| Algorithm | parameter increase per task ↓ | FLOPs Increase(%)↓ |
|---|---|---|
| GAN Memory | 4.21M | 15.7 |
| CAM-GAN | 1.52M | 23.32 |
| **StyleCL** | **1.08M** | **4.1** |

Table 1: Comparison of our approach against the baselines GAN Memory and CAM-GAN with respect to parameter increase per task and percentage increase in FLOPS. Note that both the baselines also store the parameters for each task.

Tab. 2 summarizes the quantitative results, which shows that StyleCL outperforms all other baselines for most cases in terms of FID, Density, and Coverage metrics. Furthermore, Tab. 1 shows the amount of parameter reduction and percentage increase in FLOPs. StyleCL has relatively lower per-task parameter requirements compared to other methods, even though it does not have efficient adaptation modules like CAM-GAN. This reduction in the number of parameters can be attributed to the fact that StyleCL achieves continual adaptation using a combination of feature transformation and latent space modulation. The latent space modulation requires a lesser number of parameters, while still allowing the generation of some features of the target manifold. While MerGAN has no increase in parameter or FLOP count, it comes at the expense of decreased generation quality on earlier tasks, which does not occur in our no-forgetting setting.

| Method | MerGAN | | | GAN Memory | | | CAM-GAN | | | StyleCL | | |
|---|---|---|---|---|---|---|---|---|---|---|---|---|
| Dataset/Metric | FID | D | Cov | FID | D | Cov | FID | D | Cov | FID | D | Cov |
| **Flowers** | 45.14 | 0.6 | 0.49 | 23.97 | 0.73 | 0.71 | 23.38 | **0.89** | 0.71 | **18.48** | 0.67 | **0.77** |
| **LSUN Church** | 31.41 | 0.56 | 0.18 | 37.9 | 0.30 | 0.11 | 24.25 | 0.20 | 0.17 | **17.36** | **0.59** | **0.41** |
| **LSUN Cat** | 53.52 | 1.10 | 0.20 | 53.22 | 0.86 | 0.32 | 52.59 | 0.62 | 0.22 | **34.43** | **1.15** | **0.41** |
| **Brain MRI** | 78.80 | 0.16 | 0.29 | 45.78 | 0.32 | 0.55 | 31.26 | 0.18 | 0.77 | **29.42** | **0.38** | **0.82** |
| **Chest X-Ray** | 58.51 | 0.13 | 0.11 | 58.82 | 0.23 | 0.3 | **24.81** | 0.36 | 0.73 | 25.83 | **0.55** | **0.75** |
| **Anime** | 39.83 | 0.35 | 0.09 | 16.20 | **0.63** | 0.38 | 21.52 | 0.50 | 0.27 | **12.38** | 0.62 | **0.39** |

Table 2: Comparison of the performance of StyleCL, CAM-GAN, GAN-Memory, and MerGAN on six tasks using FID (lower is better), Density (D) (higher is better), and Coverage (Cov) (higher is better). The tasks are listed along the rows and methods are listed in the columns.

## 4.4 RESULTS ON PERCEPTUALLY SIMILAR TASKS

In order to evaluate the forward transfer capability of StyleCL, we consider six varieties of butterflies from ImageNet to create a sequence of perceptually similar generation tasks, $\mathcal{X}^1$ to $\mathcal{X}^6$. We consider 2 scenarios : (a) StyleCL that enables forward transfer by considering the generator of the most similar previous task, and (b) StyleCL with parameter sharing only with the base task $\mathcal{G}^1$ (without forward transfer). Tab. 3 summarizes the results for both scenarios. We observe improved performance on most datasets for scenario (a) compared to scenario (b), confirming the benefit of forward transfer,

---

[1] https://github.com/jayleicn/animeGAN

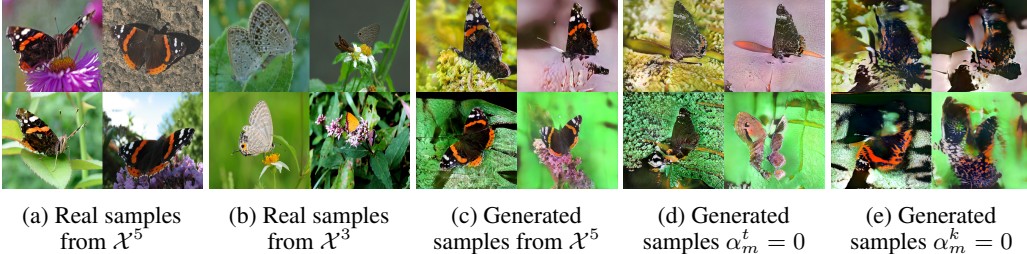

| (a) Real samples from $\mathcal{X}^5$ | (b) Real samples from $\mathcal{X}^3$ | (c) Generated samples from $\mathcal{X}^5$ | (d) Generated samples $\alpha_m^t = 0$ | (e) Generated samples $\alpha_m^k = 0$ |

Figure 3: **Qualitative illustration of forward transfer in StyleCL**: Fig. 3a and Fig. 3c corresponds to real and generated samples from current task $\mathcal{X}^5$. StyleCL employs feature adaptors from previous tasks (samples of which are shown in Fig. 3b) to generate shared features across tasks (Fig. 3d). Meanwhile, it utilizes newly added feature adaptors exclusively for the unique features of the current tasks (Fig. 3e).

inherent in our method. Also, the amount of knowledge that could be reused varies (positive or negative forward transfer) which leads to varying degrees of improvement. As observed from Tab. 3 in case of $\mathcal{X}^3$, $sim(t, k) < 0$ indicates potential negative transfer and hence when the model is forced to reuse the most similar task, it results in a performance drop. This empirically validates our characterization of the nature of forward transfer by using $sim(t, k)$. In such cases, we prevent negative forward transfer by avoiding parameter reuse from the most similar task

To qualitatively evaluate the forward transfer capability of our approach, StyleCL, we train it on dataset $\mathcal{X}^5$ shown in Fig. 3a using the generator of the most similar previous task, $\mathcal{X}^3$ whose samples are shown in Fig. 3b. The generated samples are illustrated in Fig. 3c. To analyze the individual contribution of current and previous feature adaptors in StyleCL, we separately disable their individual contribution by setting $\alpha_m^t$ and $\alpha_m^k$ to 0 in equation 3. The corresponding generated samples are illustrated in Fig. 3d

|  | $\mathcal{X}^2$ | $\mathcal{X}^3$ | $\mathcal{X}^4$ | $\mathcal{X}^5$ | $\mathcal{X}^6$ |
|---|---|---|---|---|---|
| StyleCL without transfer | 28.96 | **35.68** | 20.90 | 27.51 | 31.84 |
| StyleCL with transfer | **21.86** | 37.38 | **18.87** | **23.18** | **31.00** |
| $sim(t, k)$ | 0.59 | -0.02 | 0.21 | 0.35 | 0.37 |

Table 3: Comparison of StyleCL with and without forward transfer on a stream of perceptually similar datasets denoted as $\mathcal{X}^i$, $2 \leq i \leq 6$. Results are reported in terms of FID, where lower values indicate better performance.

and Fig. 3e. Our results show that $\phi_m^3$ is reused to capture shared characteristics of $\mathcal{X}^5$ and $\mathcal{X}^3$, such as shape and background (as seen in Fig. 3d), whereas newly introduced feature adaptors $\phi_m^5$ capture features unique to $\mathcal{X}^5$, such as the orange colour of the wings (as seen in Fig. 3e). These findings confirm that StyleCL enables forward transfer by reusing knowledge from previous tasks.

## 4.5 OVERCOMING TASK ID CONSTRAINTS WITH STYLECL

To provide empirical evidence that StyleCL inherently segregates datasets within a task, we created two distinct tasks: one that combines Flowers and Brain MRI images and another that merges Anime and LSUN Church images. For each task, we randomly sampled from the individual datasets to simulate a balanced mixture. We initialized StyleCL with two sets of dictionary vectors and feature adaptors, one for each

| Task | Datasets | FID |
|---|---|---|
| Flowers & Brain-MRI | Flowers | 24.38 |
|  | Brain-MRI | 35.22 |
| LSUN Church & Anime | LSUN Church | 23.48 |
|  | Anime | 14.63 |

Table 4: Performance of StyleCL on the task ID free setting on two data mixtures.

dataset in a task. After completing training on this mixed dataset, we evaluated the generation quality by generating samples from each component distribution using the corresponding dictionary and

feature adaptor pair. The results are presented in Tab. 4. As observed from Tab. 4, StyleCL maintains high generation quality even in the absence of task ID information.

## 5 ABLATIONS AND ANALYSIS

### 5.1 ANALYSIS OF LEARNED LATENT SUBSPACES

The learned latent dictionary for a task characterizes its position within the latent space and plays a crucial role in identifying both the most similar tasks and in preventing negative forward transfer.

To validate the effectiveness of these learned latent vectors in capturing the semantics of each task, we present t-SNE visualizations of the latent vectors for a selected set of tasks. In particular, we aim to demonstrate that latent vectors associated with similar tasks are clustered closely together while remaining distinct from those associated with dissimilar tasks. To illustrate this, we generate t-SNE visualizations for the latent vectors of two distinct Butterfly datasets Sec. 4.4 and a perceptually different task Brain-MRI Sec. 4.3. The resulting t-SNE visualization is presented in Fig. 4. As observed in Fig. 4, latent vectors of different tasks forms clusters in latent space with latent vectors of semantically similar task lying close together.

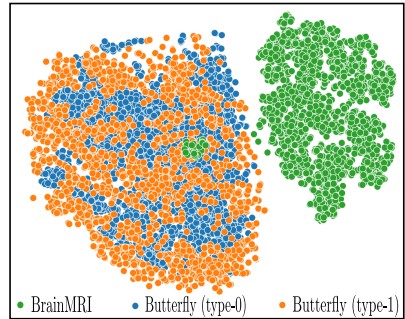

Figure 4: t-SNE visualization of latent vectors of similar and dissimilar tasks.

### 5.2 EFFECT OF GENERATOR INITIALIZATION

The generator $\mathcal{G}^1$ is obtained by training on $\mathcal{X}^1$ and shares parameters with all subsequent tasks. To analyze this initialization's impact on StyleCL, we experiment by initializing the generator with weights trained on Brain MRI and ImageNet datasets. We evaluate StyleCL's performance on a data stream consisting of CelebA-HQ, Flowers, LSUN Church, and Chest X-Ray, using these different initializations. The results in Tab. 5 demonstrate a significant performance boost when the generator is initialized with weights from a diverse dataset like ImageNet-1K, compared to a more domain-specific base task like Brain MRI. This suggests our method benefits from initial weights trained on a diverse dataset.

|  | CelebA-HQ | Flowers | LSUN Church | Chest X-Ray |
|---|---|---|---|---|
| Brain-MRI | 22.82 | 31.98 | 55.45 | 29.93 |
| ImageNet | **15.86** | **14.25** | **11.71** | **23.54** |

Table 5: This table presents the performance comparison of StyleCL (measured by FID) on four different datasets(along columns), using various generator initialization methods(along rows).

## 6 CONCLUSIONS AND FUTURE WORK

We introduce StyleCL, a lightweight expansion-based approach for generative continual learning with StyleGAN. Unlike prior methods of transforming feature maps or weights, we harness StyleGAN's latent space for continual learning. For each new task, we learn a latent subspace via dictionary learning in the $\mathcal{W}^+$ space and a feature adaptor. The proposed method requires less computational and memory overhead than contemporary methods while ensuring similar or better performance. Our future work involves (i) Extending our method to various architectures and generative models, including Diffusion models. (ii) Improving continual learning by sharing dictionaries and exploring common subspaces. (iii) Enhancing StyleCL performance in task ID-free settings with semantically similar datasets.

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

# Supplimentary Material

## CONTENTS

## A  OVERVIEW

Recall from our main manuscript that we present a lightweight expansion-based method called StyleCL for generative continual learning with StyleGAN. Unlike previous approaches that perform a transformation on either the feature maps or the weight parameters, we efficiently leverage the latent space of StyleGAN. The proposed method ensures comparable or better performance while using less computational and memory overhead than earlier approaches.

We include additional details in the supplementary material to keep the overall manuscript self-contained. The details comprise implementation details, additional experiments, additional ablation studies, and insights into our results, which could not be self-contained due to restrictions in the length of the main manuscript.

## B  RESULTS

### B.1  IMPLEMENTATION DETAILS

We adapted the author's Pytorch implementation of StyleGAN2-ADA for all our experimentations. All the datasets were scaled to 256 x 256 resolution. The number of dictionary vectors per style block was chosen as 16 for all our experiments. All other hyperparameters were the same as those of the original implementation of StyleGAN2-ADA.

The calculation of FID (Fréchet Inception Distance) involves utilizing the image embeddings extracted by the InceptionV3 model. On the other hand, Density and Coverage metrics rely on the embeddings provided by the VGG-16 model.

For lifelong classification experiments, we used the same set of hyperparameters and configurations as described in CAM-GAN.

## B.2 Additional Results

We could only display a limited number of samples generated using StyleCL in the main paper due to space constraints. Hence, we present additional samples to strengthen the qualitative evaluation of the proposed method in Fig. 12, Fig. 13, and Fig. 14. We observe that the qualitative analysis supports the quantitative metrics in terms of FID scores, Density, and Coverage metrics. We also provide an extended version of Fig. 2 presented in the main paper to provide a more comprehensive qualitative comparison of the StyleCL with other baselines Fig. 5. We further provide the absolute value of parameters (in Millions(M)), FLOPs (measured in gigaFlops(G)) and additional information in Tab. 6

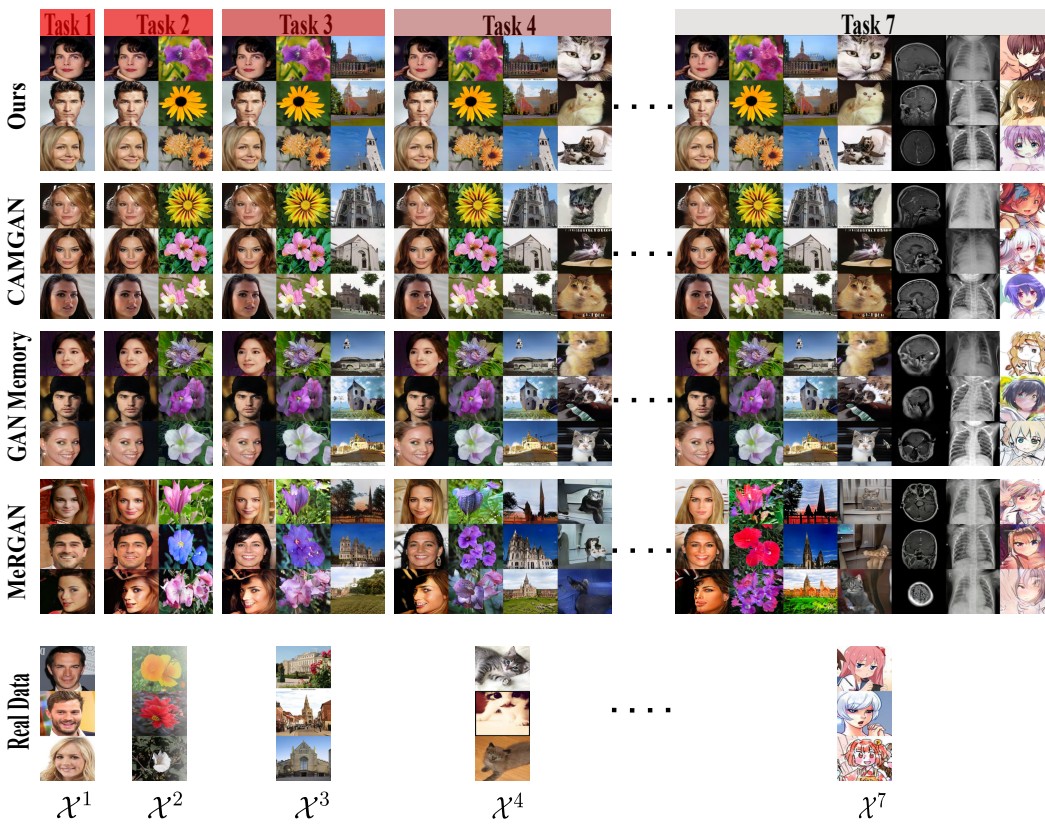

Figure 5: Qualitative results of StyleCL for all six tasks including the base task. Each row corresponds to generated images from a given task. Here $\mathcal{X}^1$ represents the base task (CelebA-HQ dataset) and $\mathcal{X}^2 \dots \mathcal{X}^7$ are further tasks (Oxford 102 Flowers, LSUN Church, LSUN Cat, Brain MRI, Chest-X Ray and Anime).

| Algorithm | Parameters (M) | FLOPs (G) | Parameters Increase per Task (M) ↓ | FLOPs Increase (%) ↓ |
|---|---|---|---|---|
| MerGAN | 24.94 | 14.00 | - | - |
| GAN Memory | 29.15 | 16.19 | 4.21 | 15.7 |
| CAM-GAN | 26.46 | 17.29 | 1.52 | 23.32 |
| **StyleCL** | 26.02 | 14.57 | **1.08** | **4.1** |

Table 6: Comparison of our approach against the contemporaries GAN Memory Cong et al. (2020), CAM-GAN Varshney et al. (2021), and MerGAN with respect to Parameters, FLOPs, Parameter reduction factor, and percentage increase in FLOPs.

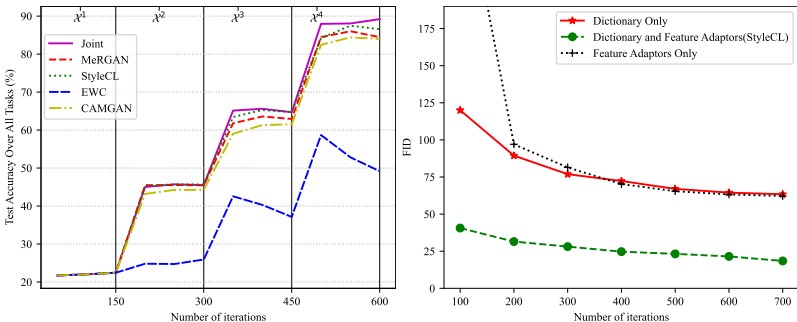

Figure 6: (Left) Classification accuracy on the lifelong classification problem See: Sec. 4. (Right) Quantitative evaluation of the contribution of different components of StyleCL for image generation.

### B.3   LIFELONG CLASSIFICATION

The conditional variant of StyleCL can serve as a generative replay buffer within discriminative continual learning. To gauge its effectiveness in aiding the classifier model during lifelong learning, we conduct tests within the generative replay paradigm. Specifically, we explore conditional data generation in generative replay experiments, applying it to the class-incremental setup designed for classification tasks. Our approach involves selecting images of fish, birds, snakes, and dogs from the ImageNet dataset and creating 4 distinct streams of tasks as in CAM-GAN. Each task is framed as a 6-classification problem, such as identifying 6 different types of birds in the bird task. After completing each task $t$, the learned classifier is required to accurately classify all the categories observed so far, up to and including task $t$. The quantitative comparison of StyleCL with regularization-based baselines EWC Lee et al. (2017) and Replay-based methods such as MerGAN, and CAM-GAN for lifelong classification tasks is demonstrated in Fig. 6 (Left). It can be observed that the StyleCL performs better compared to the baselines when used for lifelong classification tasks, because of its superior generative capability. The method that closely competes with StyleCL for lifelong classification tasks is MerGAN. However, the performance of MerGAN degrades as the task sequence increase since MerGAN learns new task at the expense of forgetting previous tasks. Generated samples using StyleCL are given in Fig. 7. The upper bound for classification performance (denoted as Joint) is obtained by using all the data till the current task to train the classifier. The performance of StyleCL is seen to be close to this upper bound.

### B.4   INTERPOLATION

#### B.4.1   INTERPOLATION WITHIN A TASK

The smoothness of the learned latent subspace is one of the properties enforced in StyleCL. We conduct a qualitative evaluation to support this claim which is demonstrated in Fig. 15 and Fig. 16 for Anime and Flowers datasets respectively. We sample two vectors randomly from the latent subspace learned for each dataset and perform linear interpolation to generate images corresponding to these vectors along the interpolation path. It can be observed that the linear interpolation in latent space produces a smooth transition in pixel space as well, demonstrating the smoothness of the latent subspace that has been learned.

#### B.4.2   INTERPOLATION ACROSS TASKS

We perform linear interpolation among various task generation processes on randomly sampled vectors from the latent subspaces learned for two distinct datasets. We perform the analysis for dataset pairs, namely (Brain MRI, Chest X-Ray) and (Flowers, Anime) which is demonstrated in Fig. 17 and Fig. 18 respectively. The linear interpolation is performed on the latent vectors as well as the task-specific parameters, while the other parameters remain fixed. As observed in Fig. 17 and Fig. 18, facial artifacts arise in the generated images when moving away from learned latent subspaces, and the image quality is drastically degraded. This could be attributed to the fact that the generation process is influenced by the parameters of the base task, CelebA-HQ.

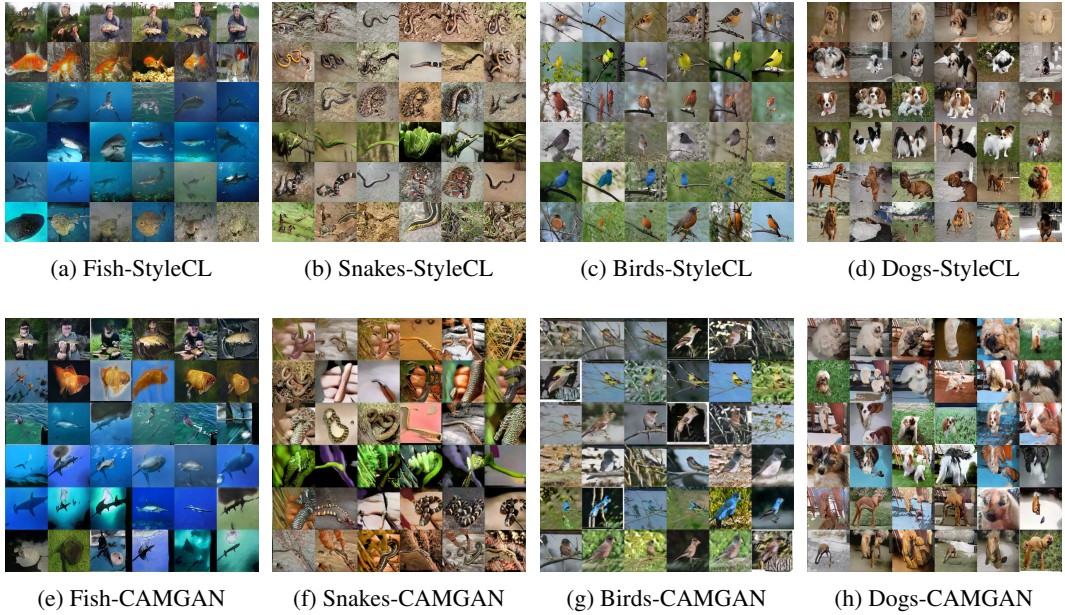

| (a) Fish-StyleCL | (b) Snakes-StyleCL | (c) Birds-StyleCL | (d) Dogs-StyleCL |
| --- | --- | --- | --- |
| (e) Fish-CAMGAN | (f) Snakes-CAMGAN | (g) Birds-CAMGAN | (h) Dogs-CAMGAN |

Figure 7: **Conditional Generation Qualitative results**: Randomly generated samples by StyleCL and CAM-GAN for lifelong classification. In each image, each row represents a class, six classes per task.

## C  ADDITIONAL ANALYSIS AND ABLATION STUDIES

In the main text, we have discussed a few analyses and ablations, in this section, we provide additional results and ablation studies for interested readers.

### C.1  EFFECT OF DIMENSIONALITY OF LEARNED LATENT SUBSPACE

To investigate the impact of the dimensionality of the learned latent space on the quality of generated images, we conducted experiments with StyleCL using only the latent subspace, varying the number of dictionary vectors. The results of this analysis are depicted in Fig. 8. Notably, the generation quality exhibits a remarkable improvement as the dimensionality of the latent subspace increases, till an optimal point. However, beyond this optimum, the generation quality starts to degrade, as illustrated in Fig. 8. We hypothesize that the additional dimensions introduce noise that interferes with the original latent vector, affecting the generation quality.

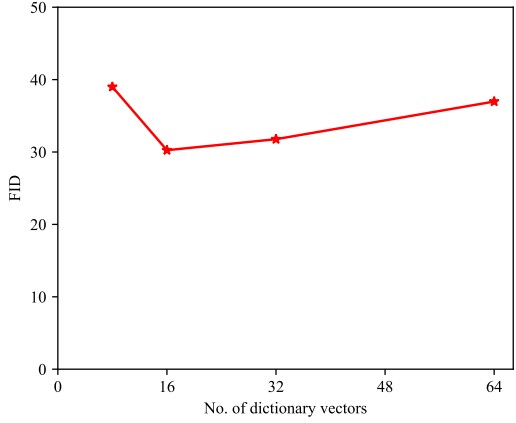

Figure 8: Quantitative ablation of the effect of dictionary size on the generation quality for Anime dataset.

### C.2  ABLATIONS OVER LATENT DICTIONARY AND FEATURE ADAPTORS

We set up ablations to understand the contributions of individual modules of our method, namely learned latent dictionary and feature adaptors. We train StyleCL with each of these modules alone and evaluate them both qualitatively (Fig. 9) and quantitatively (Fig. 6) (right). When examining Fig. 6 (right), we notice that when either of these modules is used alone, the FID is higher compared to that of StyleCL. This is further confirmed qualitatively in Fig. 9. As observed in Fig. 9, the results

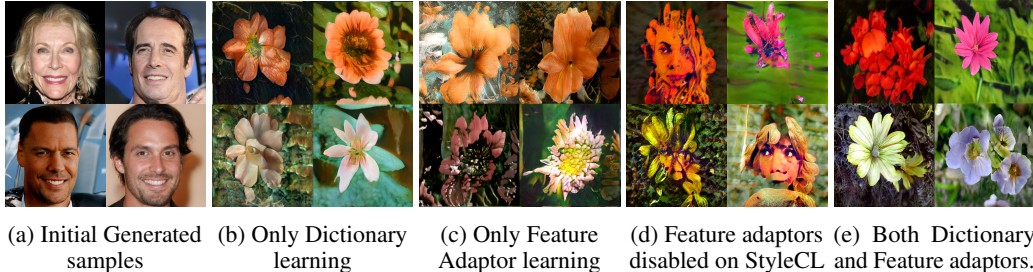

(a) Initial Generated samples    (b) Only Dictionary learning    (c) Only Feature Adaptor learning    (d) Feature adaptors disabled on StyleCL    (e) Both Dictionary and Feature adaptors.

Figure 9: **Ablation over feature adaptors and Dictionary**: Initially, the generated samples (Fig. 9a) corresponds to that of previous tasks. Learning the dictionary alone (Fig. 9b) or learning the feature adaptors alone (Fig. 9c) leads to the generated samples capturing some characteristics of the incoming data whereas combining them together (StyleCL) generates artifact-free samples (Fig. 9e). Additionally, when feature adaptors are disabled during inference in StyleCL, the generated samples (Fig. 9d) exhibit certain captured characteristics such as color and background, similar to the generated samples shown in Fig. 9e.

of dictionary learning (Fig. 9b) and feature adaptor learning (Fig. 9c) have inferior generation quality compared to the results of both modules combined (Fig. 9e).

Moreover, in order to investigate the impact of dictionary learning on StyleCL during inference, we analyze a trained model that incorporates both feature adaptors and dictionary learning. Specifically, we nullify the features obtained from the current feature adaptor while keeping the remaining components intact. The resulting generated samples, depicted in Fig. 9d, exhibit certain shared characteristics, such as colour and background, observed in the final generated samples produced by StyleCL (Fig. 9e). This finding validates our initial hypothesis, highlighting the effectiveness of StyleCL in capturing the target data manifold with a reduced parameter count, primarily due to the incorporation of dictionary learning. Additionally, in Figure 10, we present samples generated exclusively using dictionary vectors.

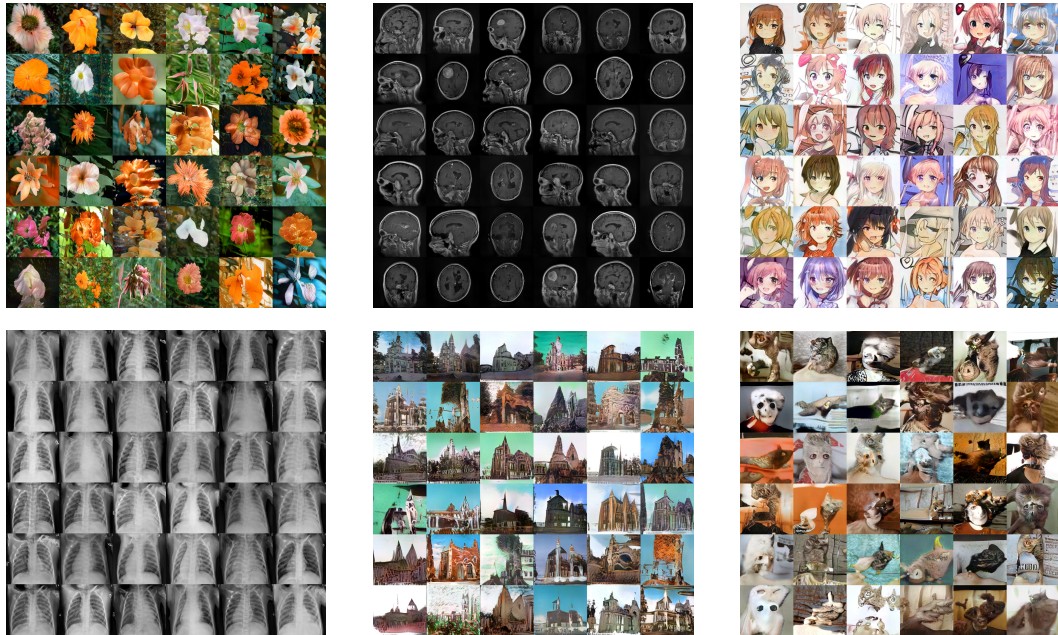

Figure 10: Generated samples by dictionary learning alone in StyleCL.

### C.3 Dictionary Learning on WGAN

Our method of learning task-specific dictionary vectors can be extended to other GAN architectures as well. In this regard, we applied it to WGAN, which does not have an extended latent space like StyleGAN. To overcome this limitation, we constructed a pseudo-extended latent space for WGAN and learn dictionary vectors over it. The linear combination of these dictionary vectors performs channel-wise Adaptive Instance Normalization (AdaIN) operation on the output of the activation layer in the ResNet block of WGAN.

AdaIN operation is performed using Equation 8 where $w_1$ and $w_2$ are linear combinations of different dictionary vectors as shown in Equation 9. Here, $\mathbf{z}_m = [z_{m1}, \ldots, z_{mK}] \sim \mathcal{N}(0, \mathbf{I})$, and $\mathbf{b}_m$ denotes the bias term similar to StyleCL.

To alter the distribution generated by the WGAN from the base task to our desired task, we perform the above-mentioned AdaIN operation on the output (x) of every activation block in the ResNet network of WGAN generator. This operation modifies the mean and variance of the output by using $\sigma(w_1)$ and $\mu(w_2)$ which we get passing $w_1$ and $w_2$ through learnable fully connected layer.

Figure 11 presents the generated samples obtained by applying our method to datasets such as Anime, Chest X-ray, and Flowers using WGAN as the base architecture with CelebA as the base task. These samples demonstrate the proposed method could be adapted to other GAN architecture as well to some extent.

$$\text{AdaIN}(x, w_1, w_2) = \sigma(w_1) * x + \mu(w_2) \tag{8}$$

$$\mathbf{w}_m = z_{m1} \, \mathbf{u}_{m1} + z_{m2} \, \mathbf{u}_{m2} \ldots + z_{mK} \, \mathbf{u}_{mK} + \mathbf{b}_m \tag{9}$$

## D  Limitations and Broad Impact

Our method finds its utility in the continual learning of lightweight generative models. This is useful in porting generative models onto to edge-devices with effective generation on streams of datasets. Currently, our method has a limitation in that it can operate only on the StyleGAN architecture. We also observed that the task ID-free setting is sensitive to the semantic similarity of datasets.

## E  Reproducibility

To facilitate reproducibility, we are attaching the code along with supplementary material as a zip file. The necessary requirements file is also attached with it. We intend to release a more user-friendly version of the code publicly along with the pre-trained models post-acceptance. All the datasets used are publicly available.

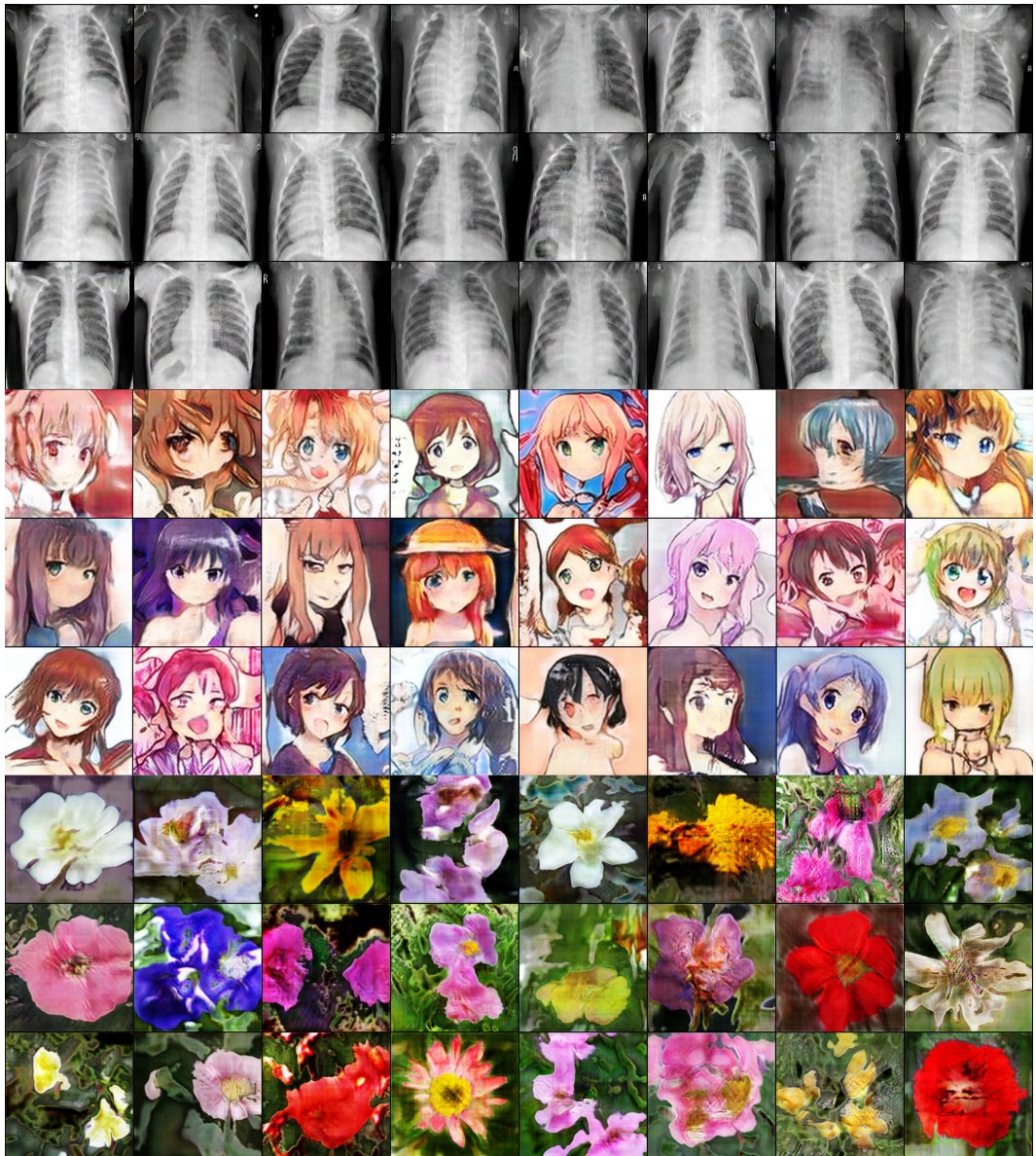

Figure 11: Qualitative results when learning dictionary vectors for WGAN architecture. **Rows:1-3**: Chest X-Ray **Rows:4-6**: Anime **Rows:6-9**: Flowers.

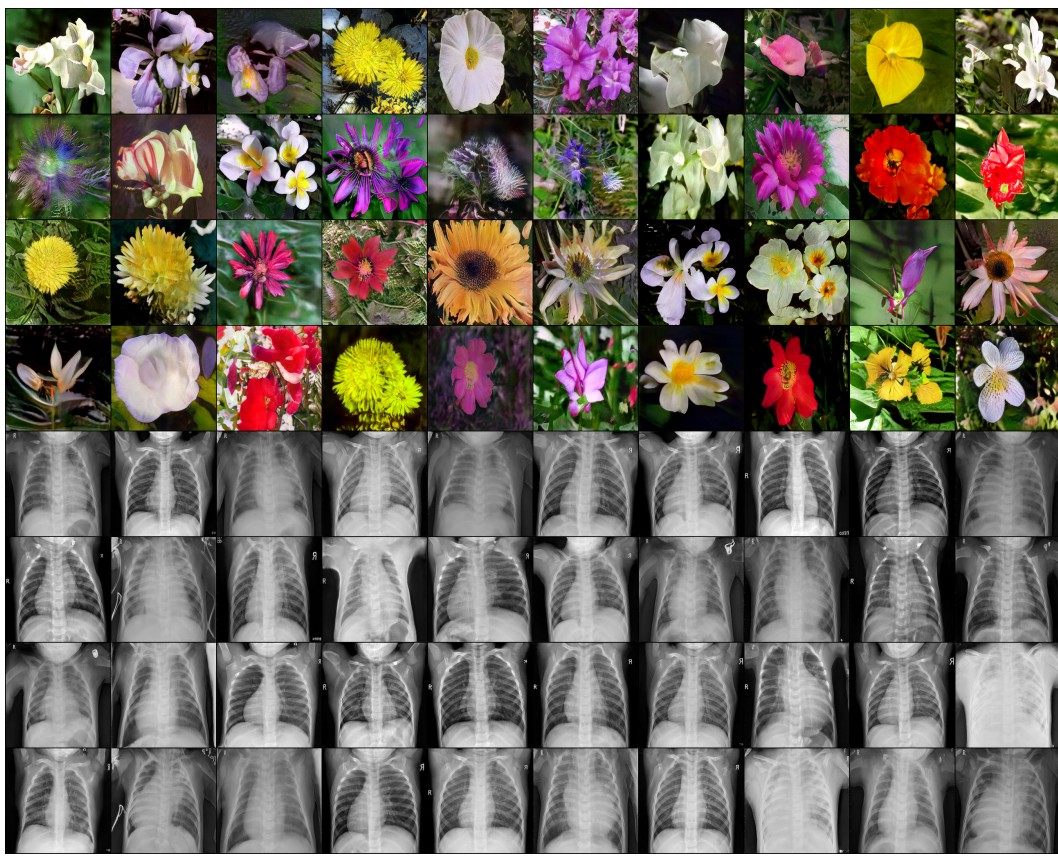

Figure 12: Randomly generated images using StyleCL. **Rows:1-4**: Flowers **Rows:5-8**: Chest X-Ray. The generated images show high similarity to real data with the exception of some finer details and artifacts.

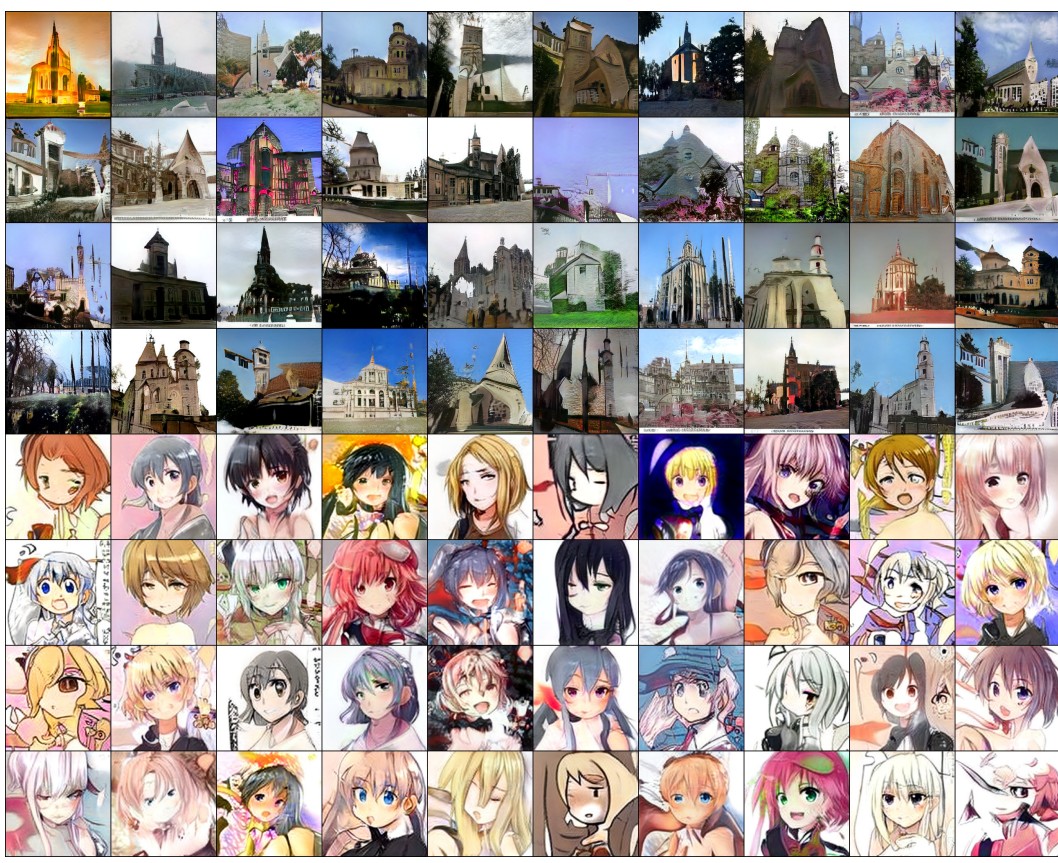

Figure 13: Randomly generated images using StyleCL. **Rows:1-4** LSUN Church **Rows:5-8** Anime. The generated images show high similarity to real data with the exception of some finer details and imperceptible artifacts.

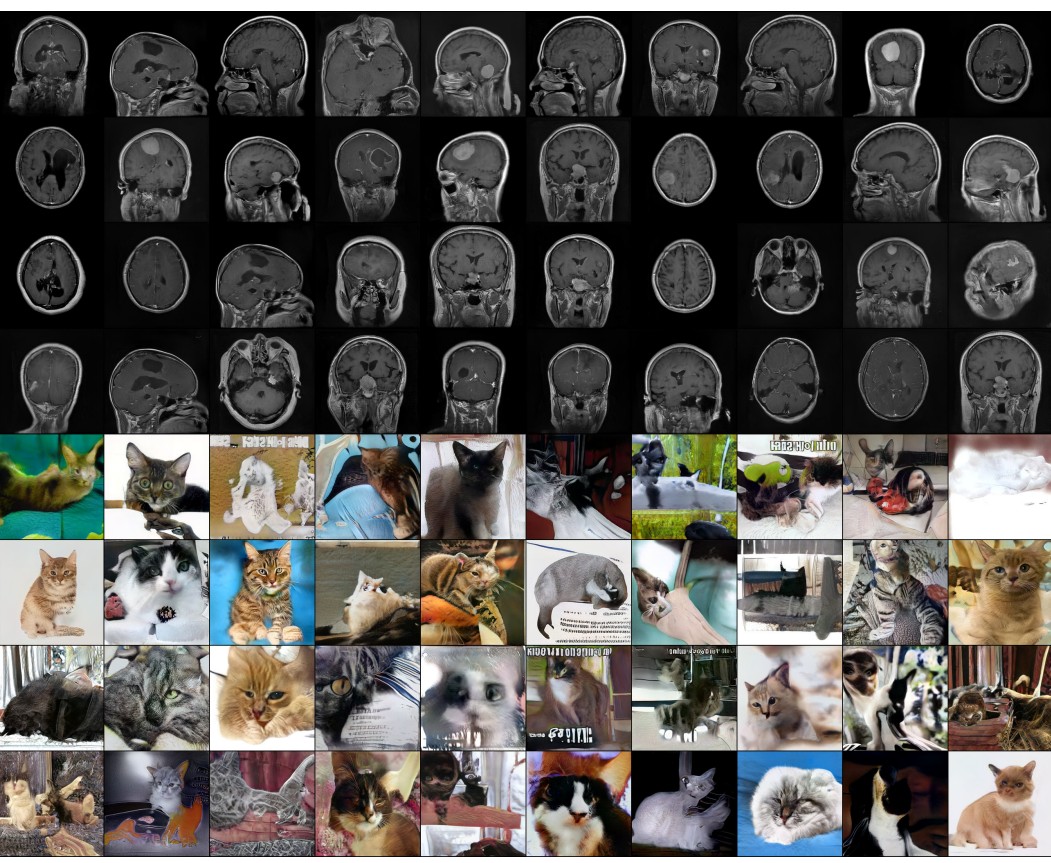

Figure 14: Randomly generated images using StyleCL. **Rows : 1-4**: Brain MRI **Rows : 5-8**: LSUN-Cats. The generated images show high similarity to real data with the exception of some finer details and imperceptible artifacts.

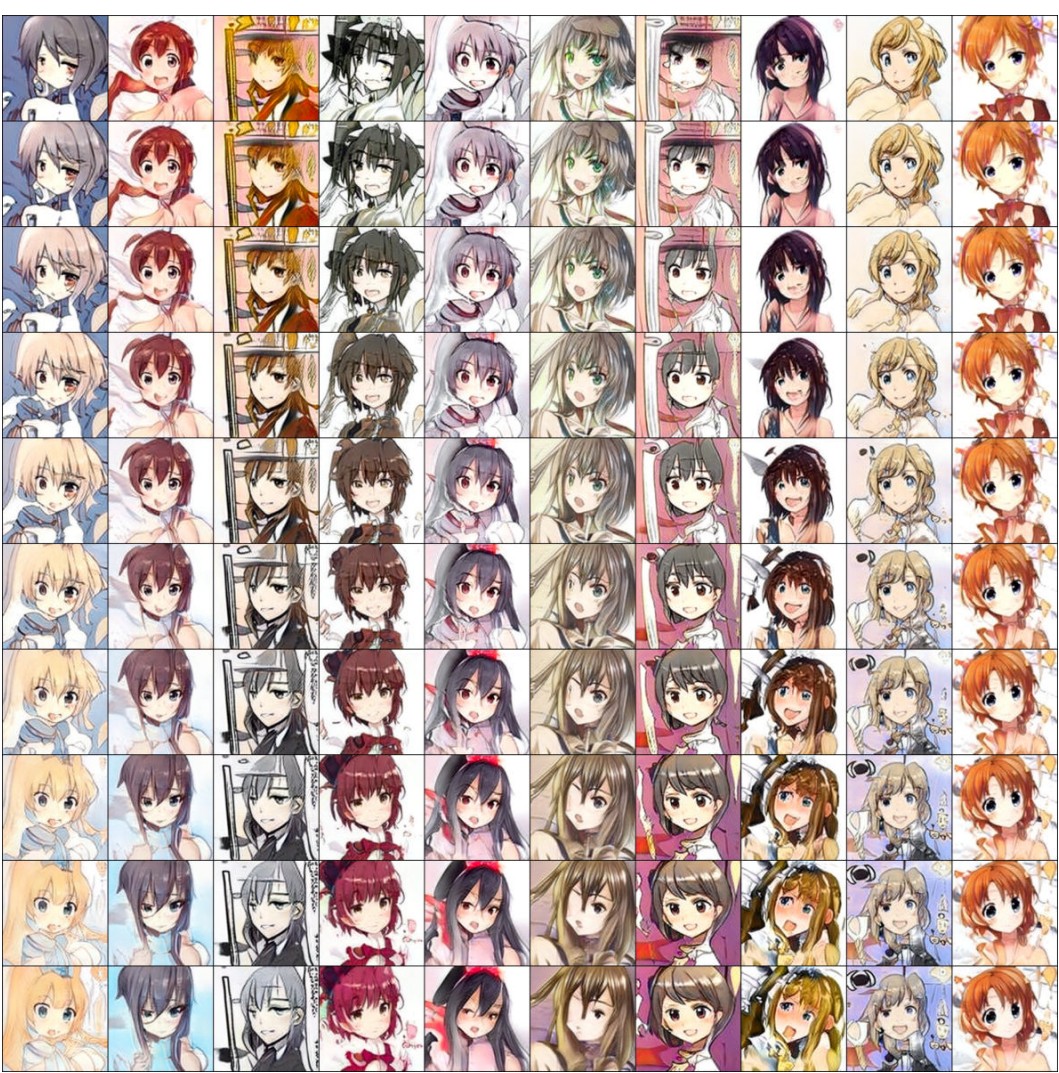

Figure 15: We perform linear interpolation ($stepsize = 0.1$) between two randomly sampled vectors from the latent subspace learned by StyleCL. Each column represents a linear interpolation where the first and last image corresponds to vectors sampled from the latent subspace of Anime.

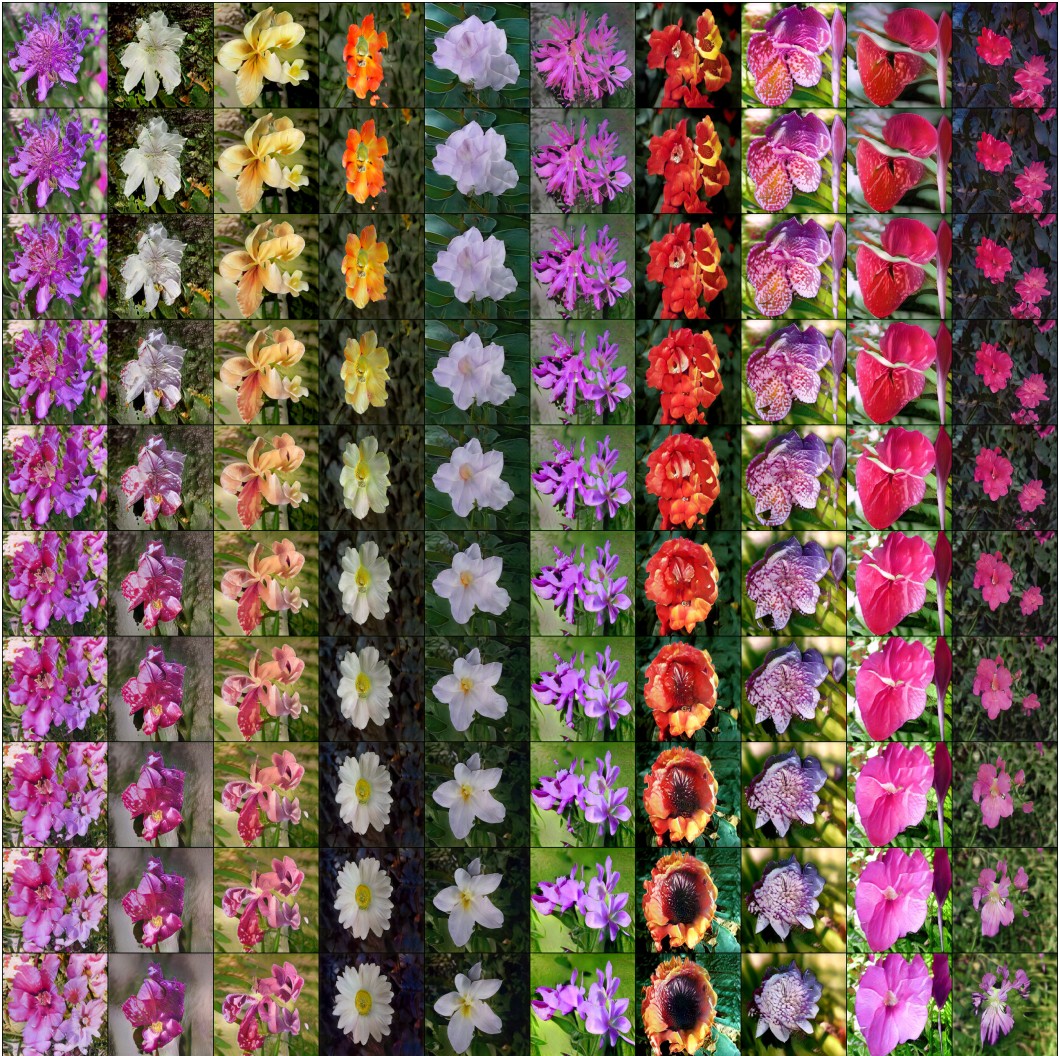

Figure 16: We perform linear interpolation ($stepsize = 0.1$) between two randomly sampled vectors from the latent subspace learned by StyleCL. Each column represents a linear interpolation where the first and last image corresponds to vectors sampled from the latent subspace of Flowers.

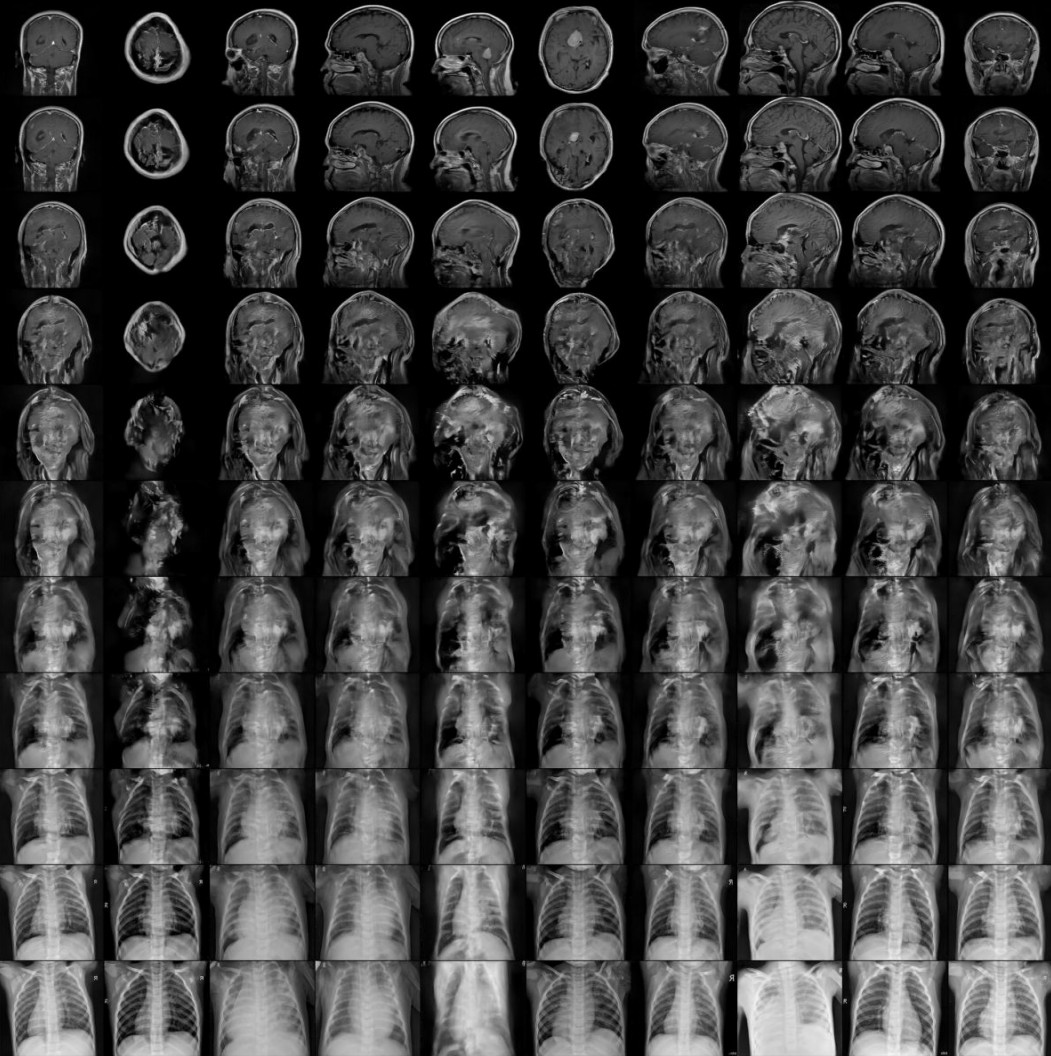

Figure 17: We perform linear interpolation ($stepsize = 0.1$) between two randomly sampled vectors from the latent subspace learned by StyleCL. Each column represents a linear interpolation where the first and last image corresponds to vectors sampled from the latent subspace of Brain MRI and Chest X-Ray respectively.

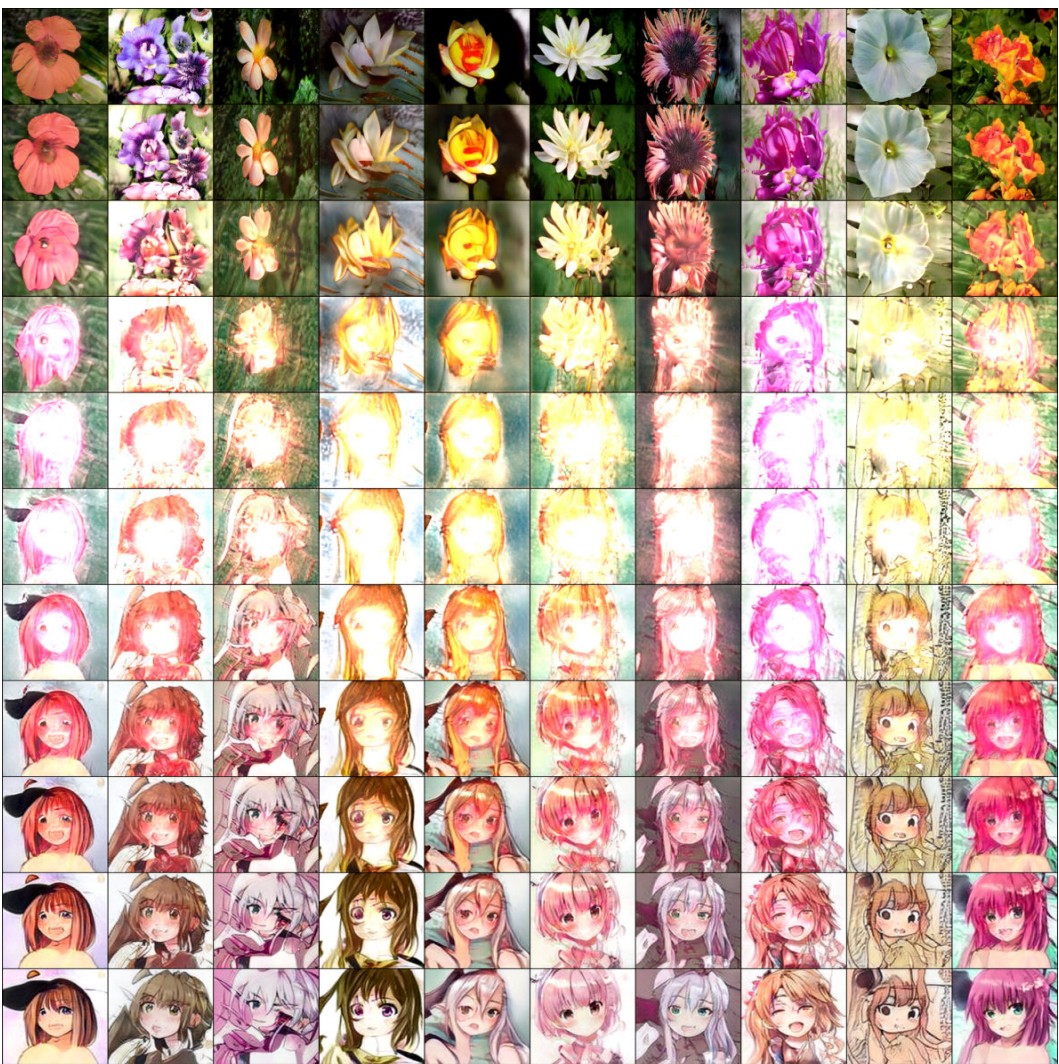

Figure 18: We perform linear interpolation ($stepsize = 0.1$) between two randomly sampled vectors from the latent subspace learned by StyleCL. Each column represents a linear interpolation where the first and last image corresponds to vectors sampled from the latent subspace of Flowers and Anime respectively.

