# OpenReview forum: "StyleCL : Latent Dictionary Learning for StyleGAN Without Forgetting"
_ICLR.cc/2024/Conference — ICLR 2024 Conference Withdrawn Submission_

### Official Review · Reviewer_HJoy · 2023-10-29

**Soundness:** 2 fair
**Presentation:** 2 fair
**Contribution:** 2 fair
**Rating:** 3
**Confidence:** 5

**Summary:**

A new method based on StyleGAN, termed StyleCL, is proposed for generative lifelong learning tasks. The StyleCL is developed based on frozen StyleGAN parameters and task-specific dictionary and feature adaptor parameters. By unifying the frozen StyleGAN parameters and the task-specific trainable parameters, the StyleCL might avoid catastrophic forgetting.

**Strengths:**

The paper is generally easy-to-follow.

The presented techniques—novel combinations of existing ones—are likely original.

**Weaknesses:**

The clarity should be improved. For example, several techniques are presented without convincing justification. See the questions for details.

The advantages and disadvantages of the proposed method when compared to existing ones are not clearly stated. See the questions for details.

**Questions:**

In Algorithm 1, for task t, there are three main iterations, right? The first one is to optimize the bias b to find the most similar previous task k; the second one is to optimize Ut, bt with Eq. (4); and the last one is to optimize Ut, bt, and ϕt using Eq. (5). How to evaluate the training efficiency of the proposed StyleCL?

On choosing the most similar previous task in Section 3.4, why can Eq. (6) be used to evaluate the similarity between the current task and previous tasks?

On preventing negative forward transfer, why can the sim(t, k) in Eq. (7) be used to signify positive or negative forward transfer? How is the formula $V_{m}^k {}^{*}U_m^t V_{m}^k$. Also, if sim(t, k)>0, then how to guarantee that optimizing the t-th objective will not lead to negative forward transfer on task k?

Section 3.5 is generally not clear. Please elaborate on how to modify Algorithm 1 to handle real-life scenarios without task IDs.

---

### Official Review · Reviewer_YAxS · 2023-10-31

**Soundness:** 3 good
**Presentation:** 3 good
**Contribution:** 3 good
**Rating:** 3
**Confidence:** 4

**Summary:**

This work propose a lifelong learning of StyleGAN without forgetting. Motivated by the finding that the latent space of StyleGAN is very versatile, the authors propose to introduce a set of dictionary vector in the latent space and a small fraction of new parameters in the weight space for each new task to be learned. Utilizing the introduced set of dictionary vectors, the authors propose a simple way to identify the most similar previous task and also characterize the nature of forward transfer between any two tasks to prevent negative forward transfer.

**Strengths:**

The proposed method utilize parameter isolation along the task stream direction, which is inherently without forgetting. The learning process have capacity of positive forward transfer for semantically similar tasks, which is expected in lifelong learning filed.  The size of newly introduced parameters for each task is smaller than existing methods.

**Weaknesses:**

The experimental results are not impressive enough. For example, the effectiveness of the positive forward transfer is only tested on a small dataset case with all tasks are positively related(all belong to the butterfly class), the forward transfer effect doesn’t show a positive relationship with the sim measure.

**Questions:**

1.	In equation(3), the activation of s^t_m includes phi^k_m and phi^t_m, it looks like a recursive definition, then the activation of s^k_m should include phi from its nearest model and the phi for itself, and so on following the recursive rules. Why all the historical nearest models are omitted in equation (3)?

2.	The right side of Equation(7) is a matrix? How to make comparison to zero?

3.	The technique about overcoming task ID constraints in lifelong learning setting is not impressive, and some related problems are not mentioned, e.g. How to set a proper number for the mixture of tasks? How to exclusively employ the feature adaptors and latent vectors since we don’t know the task ID? are the historical tasks also have no task ID?

4.	Figure 4, the t-SNE of the learned latent dictionary shows a lot of points for each task, which is confusing, since the learned latent dictionary for each task is not bigger than K, do the authors use large and different K for each task?

5.	There is no analysis about the effect of different dictionary size K on the performance? Which K is used in the experiments? How to select K for a new dataset?

6.	Table 2, are the backbone for all the compared methods styleGAN2?  Why the performance of GAN Memory here looks much worse than the original paper, where the FID are all below 20.

---

### Official Review · Reviewer_3wXE · 2023-10-31

**Soundness:** 2 fair
**Presentation:** 2 fair
**Contribution:** 2 fair
**Rating:** 5
**Confidence:** 2

**Summary:**

This paper considers the problem of continual learning for generative adversarial networks. Given a pretrained StyleGAN, this paper proposes to learn a set of dictionary vectors for each new task and shared feature adaptors. Experimental results show both qualitative and quantitative evaluation of the proposed method on image datasets.

**Strengths:**

- Continual learning of generative models is an interesting topic but less explored.

- The proposed method seems to be effective and better than previous methods according to Table 2.

- The code is provided to ensure reproducibility.

**Weaknesses:**

- The proposed method heavily relies on the StyleGAN, so it might not be generalizable to other types of GANs or generative models.

- The design choice and/or validity of the proposed method is generally not justified. Here I provide some examples, not exhaustive:
1. There might be many options other than having the MxdxK dictionary per task. For example, some or all vectors might be shared across stages or tasks.
2. The necessity of (shared) feature adaptor is not justified. Including this, authors could do ablation studies to justify the effectiveness of each proposed module.
3. The claim that "(the set of bias vectors) captures the relative position of the learned latent subspace in the W^+ space." is not justified.

- There is no comparison with baselines in experiments starting from Section 4.4, so not sure if the proposed method/module is really effective. There might be suboptimal or even redundant design choices.

- The paper is overall not well written. The aspect ratio of figures is awkward, e.g., Fig 1 and 2. There are many citation format errors. Please distinguish the usage of \citep and \citet.

- Metrics used in Table 2 are never explained, so it is hard to understand what is going on without reading references.

- The reference section requires thorough proofreading, as there are many incomplete/inaccurate references. For example, many references miss the name of the published venue or they are in inconsistent format, e.g., "In Proceedings of the IEEE Conference on Computer Vision and Pattern Recognition (CVPR)" vs. "In Proc. CVPR"

**Questions:**

Please address concerns in Weaknesses.